# mmSafe: A Voice Security Verification System Based on Millimeter-Wave Radar

**DOI:** 10.3390/s22239309

**Published:** 2022-11-29

**Authors:** Zhanjun Hao, Jianxiang Peng, Xiaochao Dang, Hao Yan, Ruidong Wang

**Affiliations:** 1School of Computer Science and Engineering, Northwest Normal University, Lanzhou 730070, China; 2Gansu Province Internet of Things Engineering Research Centre, Northwest Normal University, Lanzhou 730070, China

**Keywords:** security privacy, millimeter-wave, vocal cord vibration, playback attacks, authentication, text-independent

## Abstract

With the increasing popularity of smart devices, users can control their mobile phones, TVs, cars, and smart furniture by using voice assistants, but voice assistants are susceptible to intrusion by outsider speakers or playback attacks. In order to address this security issue, a millimeter-wave radar-based voice security authentication system is proposed in this paper. First, the speaker’s fine-grained vocal cord vibration signal is extracted by eliminating static object clutter and motion effects; second, the weighted Mel Frequency Cepstrum Coefficients (MFCCs) are obtained as biometric features; and finally, text-independent security authentication is performed by the WMHS (Weighted MFCCs and Hog-based SVM) method. This system is highly adaptable and can authenticate designated speakers, resist intrusion by other unspecified speakers as well as playback attacks, and is secure for smart devices. Extensive experiments have verified that the system achieves a 93.4% speaker verification accuracy and a 5.8% miss detection rate for playback attacks.

## 1. Introduction

In recent years, people have been interacting with many smart devices more and more frequently. As a core technology to support smart scenarios, voice assistants enable voice information interaction between people and many smart devices, such as Xiaomi’s voice assistant “Mi AI” [1] and intelligent voice control systems in cars [2]. Smart devices are developing more and more rapidly, bringing convenience to people’s lives but also some security and privacy issues, for example, in smart home and car scenarios, there can be an invasion of voice assistants by external speakers or playback attacks, resulting in some information in smart furniture or voice control systems in cars being tampered with to the point of causing significant verification problems. Currently, there is a great deal of anxiety about privacy security, and common voice spoofing attacks include voice conversion (VC), text-to-speech synthesis (TTS), impersonation, and playback of speech [3]. There is a long and rich history of defending against playback attacks, which is a core research problem in the field of biometric security [4]. This paper designs and develops the mmSafe system to address such problems.

Currently, voice speaker recognition mainly uses microphone sensors; when the speaker is speaking, the sound produced is propagated through the air medium and finally received by the microphone. However, speaker verification implemented by this method is vulnerable to playback attacks. This is difficult to solve with a software-based approach [5]. Moreover, the speaker is susceptible to other noises when speaking into the microphone, which can significantly reduce the accuracy of the speaker verification. Turbulence is determined by vocal cord vibrations, which is the root of the uniqueness of vocal print and has a promising application and high market value in the field of identity verification [6,7]. Therefore, to obtain the biometric characteristics of the speaker based on the vibration of their vocal cord part to achieve secure verification, there are some challenges for this verification method, as follows: (1) How to sense the fine-grained vocal cord vibration of the target object and then extract more accurate vocal print information based on the vibration signal of the vocal cord. (2) How to effectively defend against intrusion by other malicious attackers or playback attacks.

In order to address the above security issues and challenges, this paper proposes a millimeter-wave radar-based voice security verification system, mmSafe, which uses a 77 GHz frequency-modulated continuous-wave (FMCW) millimeter-wave radar as a sensor to sense the vibration of the vocal cords. First, the vibration signal of the vocal cords is obtained through the calculation of spurious wave cancellation, phase, and phase difference, as well as the design of filters suitable for the vibration frequencies of the vocal cords [90 Hz, 200 Hz] [8,9]; second, three sets of MFCC characteristics are obtained and weighted according to the vibration signal; finally, the weighted MFCCs are used in the WMHS (Weighted MFCCs and Hog-based SVM) method for security verification. The main contributions of this paper can be summarized as follows:This paper proposes a vibration signal processing method that solves the problem of clutter as well as the effects of motion and achieves an enhancement of the vocal cord vibration signal and vocal print information;This paper proposes the WMHS (Weighted MFCCs and Hog-based SVM) method, which converts the speaker verification problem into a binary classification problem by combining weighted MFCCs with a HOG feature-based SVM to improve the verification accuracy;This paper develops the mmSafe system, which senses speakers in a non-contact and more fine-grained way, and which has been proven to be effective in verifying speakers and resisting playback attacks through extensive experiments, achieving a 93.4% speaker verification accuracy and a 5.8% miss detection rate for playback attacks.

## 2. Related Work

### 2.1. Speaker Verification with Microphone

The most popular speaker verification is currently sensed through microphones, and Roberto Font et al. [10] have carried out a significant amount of work on acoustic feature extraction, and their study found that mute segment cancellation reduces detection accuracy when preprocessing audio for playback attack detection; Cepstral Mean Normalization (CMN) of acoustic features [11] helps improve system performance. Chen et al. [12] proposed a single-channel speech separation deep learning framework to achieve multiple-speaker separation. Liang et al. [13] achieved multi-channel speaker verification by using ad hoc microphone arrays.

### 2.2. Speaker Verification with Millimeter-Wave Radar

Unlike microphones, a millimeter-wave radar propagates in an electromagnetic field without any other medium [14], so noise in the air cannot interfere with a millimeter-wave radar. Millimeter-wave radars are very sensitive to fine vibrations due to their short wavelength (about 4 mm) and are a promising technique for small displacement measurements. One study demonstrated the acquisition of sound vibrations in the range of 2–3 mm by a millimeter-wave radar [15], and He et al. [16] investigated a practical method for the millimeter-wave radar measurement of micron-scale vibrations. Lin et al. [17] investigated systems that provide long-range, noise-resistant, and motion robustness in public applications, addressing the sound in complex environments where the surrounding environment and large body movements have adverse effects.

### 2.3. Speaker Verification with Other Sensors

For speaker verification and to resist malicious attacks, many researchers use other sensors. For example, Liu et al. [18,19] used a camera as a sensor to capture the speaker’s lip movements for speaker verification, but the privacy of the speaker is violated by this method. Other researchers have used laryngeal microphones to achieve speaker verification by sensing the vibrations of the vocal cords [20]. Although these methods are good for verifying speakers and resisting malicious attacks, laryngeal microphones require the speaker to wear a microphone around the neck for sensing, which is very inconvenient. Zhang et al. [21] proposed smartphone-based voice sensing for contactless sensing and speaker verification, but the usage scenario of this method is very limited and can only be used in smartphones.

On the one hand, the mmSafe method developed in this paper uses only one signal of vocal fold vibration to verify the speaker compared to the methods in the literature [8]. Although the accuracy of verifying the speaker found in the literature [8] is 95.93%, the computational overhead of multimodality is larger than that of mmSafe; additionally, the signal of lip movement cannot be extracted when the speaker is wearing a mask. In the literature [22], although the overall accuracy of the verification is 93.5%, only smartphones can be used, which does not solve the problem of scenario limitation, and the accuracy rate drops to less than 20% when the speaker is more than 12 cm away from the phone for security verification, but mmSafe can be used in any smart voice device and has good applicability. As a result of these works, this paper has developed the mmSafe system, which senses the speaker’s vocal cord vibrations by using a 77 GHz FM continuous-wave (FMCW) millimeter-wave radar as a sensor that is not affected by environmental noise nor does it need to be worn on the body; moreover, the millimeter-wave radar has a more fine-grained sensing capability.

## 3. System Design

In this section, the overall structure of the mmSafe system is described, as well as the detailed workflow of each module.

### 3.1. System Overview

The structure diagram of the mmSafe system is shown in Figure 1. The system is divided into three modules in total. First, in the radar signal processing module, the raw data of the speaker’s speech are collected by the IWR1642BOOST millimeter-wave radar for signal processing, and the vibration signal of the vocal cords is finally obtained from the phase and phase difference of the signal. Next, the speaker’s features (weighted MFCCs) are extracted in the vibration feature extraction module. Finally, speaker verification is achieved by WMHS in the speaker verification module. As shown in the bottom left corner of Figure 1, only the specified speaker can be verified; other speakers as well as playback attacks cannot pass the verification.

### 3.2. Radar Signal Processing Module

In the radar signal processing module, the transmitted sinusoidal signal is assumed to be *t*(*x*) and the received sinusoidal signal is *r*(*x*):(1)$$t(x)=\sin(w_{1}t+\varphi_{1})$$
(2)$$r(x)=\sin(w_{2}t+\varphi_{2})$$

The instantaneous frequency of the *IF* signal synthesized and output by the mixer is assumed to be *IF_out_*, which is equal to the difference between the instantaneous frequencies of the two input sine signals, and the instantaneous phase is the difference between the phases of the two input signals:(3)$$IF_{out}=\sin[(w_{1}-w_{2})t+(\varphi_{1}-\varphi_{2})]$$

The frequency of the intermediate frequency (*IF*) signal is:(4)$$f_{IF}=S\tau =\frac{2Sd}{c}$$
where S is the slope of the linear FM pulse, d is the distance between the target and the radar, τ is the time delay between the transmitted signal and the received signal, and the speed of light is: $c=\frac{2d}{\tau }$.

The amplitude of vocal cord vibration is from about 0.3 to 1.5 mm due to the fine round-trip motion of the laryngeal skin during vocal cord vibration [23]. As shown in Figure 2, such a round-trip motion can be captured by the electromagnetic waves emitted by the FMCW radar and cause a phase change, so the prerequisite to obtain the vibration signal is to first calculate the phase of the speaking activity.

In the first step, because the surrounding static object clutter will affect the vibration signal, the module needs to remove the static clutter first using the average phase cancellation method, and then perform a one-dimensional fast Fourier transform on the intermediate frequency (IF) signal collected by the radar, as shown in Figure 3a, of the measured target from eight distinct distances, after obtaining the correct Rang-bin, and then find the phase when the speaker performs the speaking activity:(5)$$\varphi_{S}=2\pi f_{IF}\tau$$

The phase is shown in the upper part of Figure 3b, and by further derivation, it can be found that:(6)$$\varphi_{S}=\frac{4\pi d}{\lambda }$$
where λ is the wavelength.

In the second step, it can be seen from the top half of Figure 3b that a significant amount of phase information has jumped, and to obtain smoother phase information the phase needs to be unwrapped; the bottom half of Figure 3b represents the post-phase unwrapping. In order to enhance the phase information, the module performs a phase difference operation, and the result is shown in Figure 3c.

In the last step, the module obtains the vocal cord vibration signal of the speaker by designing a filter, and since the vocal cord vibration has a high frequency, the signal of the subtle body movements can be separated from the vocal cord vibration by a band-pass filter. Figure 3d shows the vibration signal of the vocal cords when the speaker is speaking, and it can be seen that there is speaking activity at about 3 s. Next, to further process the vibration signal, mmSafe inputs the vibration signal obtained in the radar signal processing module into the vibration feature extraction module.

### 3.3. Vibration Feature Extraction Module

The MFCC is a feature extraction method, proposed by Davies and Mermelstein around 1980, that can be good at detecting differences in speech signals [24,25,26], and since sound recognition in the human auditory system is performed by signals generated by eardrum vibrations, the MFCCs can also produce good results in vibration signal processing [27].

Figure 4 shows the diagram of the process of calculating the weighted MFCCs. There are three diagrams in the lower part of Figure 4. In the first step, the first of them is the traditional MFCC, which will be defined as “feat-MFCC”; the second diagram is obtained after the first-order difference of “feat-MFCC”, which will be defined as “defeat-MFCC”; and then the third diagram is obtained after the second-order difference of MFCC, which will be defined as “dttfeat-MFCC”. In the second step, the three sets of MFCC are weighted to obtain the weighted MFCC, which will be defined as “final-MFCC”, which is the upper right part of Figure 4, and the characteristics of the vibration signal can be seen in the box in the figure. The red box in the upper left corner of Figure 4 represents the vibration signal, while the red box in the upper right corner of Figure 4 represents the feature corresponding to the vibration signal.

The detailed steps of the module’s work are described below.

The MFCC features are extracted based on the Mel frequency, wherein the Mel frequency is related to the frequency of the vibration signal as:(7)$$\mathrm{Mel}(f)=2595\mathrm{lg}(1+\frac{f}{700})$$
where Mel(f) is the Meier frequency and f is the frequency of the vibration signal.

The entire process of obtaining MFCC parameters can be divided into the following steps: first, to strengthen the high-frequency information, the signal should be weighted, and the weighting function is as follows:(8)y(n)=x(n)−θ·x(n−1)
where y(n) is the signal after the desired weighting, x(n) is the signal before the desired weighting, and the module takes the value of θ to be 0.95.

Next is framing, which often divides the signal into 20 ms to 40 ms frames, and this module divides the signal into 25 ms frames. Then, each frame is brought into the window function, which is used as a window function for the Hamming window.

Then, the Fourier transform is performed: this module performs the Fourier transform on the signal after adding the window for each frame before deriving the power spectrum.

The next step is the Meier filter: a set of 26 triangular filters, which filter the power spectrum obtained in the previous step.

The last step is the discrete cosine transform: this module needs to conduct the operation of the discrete cosine transform on the logarithm of the energy to obtain the “feat-MFCC”. Then, the “feat-MFCC” is weighted and combined with the first-order difference and second-order difference to obtain the “final-MFCC”. Then, to verify the speaker, mmSafe inputs the final-MFCC obtained from the vibration feature extraction module to the speaker verification module.

### 3.4. Speaker Verification Module

The module converts the speaker verification problem into a binary classification problem and implements it using Algorithm 1, which has the following pseudo-code.
**Algorithm 1:** WMHS.**Input:** mfcc-final←[feat,defeat,dttfeat]**Output:** verification results  1: img.power←mfcc−final.img(255*255);
  2: G←gradient_magnitude←$\sqrt{G_{x}^{2}\left(x,y\right)+G_{y}^{2}(x,y)}$; // Calculate the total gradient value  3: Angle←$\alpha \left(x,y\right)=\tan^{-1}\frac{G_{y}\left(x,y\right)}{G_{x}\left(x,y\right)}$;  4: **function** UPDATEBINS(self,G, Angle)  5: **while** dividing the image into cells **do**  6: bins←zeros((cell_G.shape[0], cell_G.shape[1],self.bin_count));  7: **for** i = range(bins.shape[0])  8: **for** j = range(bins.shape[1])  9: bins←tmp_unit←self.bin_count; // Vote for each gradient direction  10: **end for**  11: **end for**  12: **end while**  13: **return bins;**  14: **end function**  15: block←bins.feature  16: $F^{HOG}=\left(F_{1}^{HOG},F_{2}^{HOG},\cdots ,F_{N}^{HOG}\right)$ ← block  17: clf = svm. SVC(); // training model  18: clf.fit(train_ reduction, train_ target);  19: classification function ← $f(x)=\mathrm{sgn}\left\{\left(\sum_{i=1}^{n}a_{j}{}^{*}y_{j}(p_{j}\cdot q_{i})\right)+b^{*}\right\},p\in R^{n}$  20: pred = clf.predict(test_reduction) // predict  21: **return precision**

## 4. Experimentation and Evaluation

In this section, an experimental evaluation of the mmSafe system is conducted to verify not only the overall performance of the mmSafe system but also to test the effect of different factors on the speaker verification results.

### 4.1. Experimental Setup

In this paper, we use the IWR1642BOOST millimeter-wave radar and a Xiaomi loudspeaker, model XMYX07YM, used to test playback attacks. The IWR1642 device is a single-chip millimeter-wave radar sensor operating in the frequency band of 76 to 81 GHz [28]. The following parameters are set for this radar: beat frequency sampling rate of 5 MHz, slope of 66.626 (MHz/μs), 10 Chirp in each frame, 2 ms duration of each frame, and set for 2000 frames. The radar has two transmitting antennas and four receiving antennas, and the experimental setup is shown in Figure 5, which shows that the IWR1642BOOST millimeter-wave radar and the DCA1000EVM data acquisition board are placed 20 cm in front of the measured target to obtain the raw ADC data and transfer the data to the computer through the USB data cable. When the computer receives the raw data, MATLAB is used to parse and process the data based on hardware configured with i5-11400H and 16 G memory.

The experiment comprises a total of 10 target objects, 1 designated speaker, and 9 other attackers, one of which is a Xiaomi loudspeaker. Each target faces the radar from a distance of 20 cm and speaks a word within 4 s. A total of 10 different words are collected as one set of experimental data. A total of 30 sets of experimental data are collected, 20 of which are used as the training set and the other 10 as the test set.

As shown in Figure 6, speaker verification is a dichotomous problem, in which we can see that the first category is speaker, that is, the main speaker, and the second category is other speakers and playback attacks, that is to say that, except for the main speaker, they all belong to the second category.

### 4.2. System Performance

In this subsection, we evaluate the accuracy of speaker verification and resistance to playback attacks, but also the impact of the speaker on the performance of the mmSafe system in three different dimensions: the spatial dimension, the algorithmic dimension, and the speaker state dimension, respectively.

#### 4.2.1. Spatial Dimension

In order to test the effect of the speaker on the system performance in the spatial dimension, we set up the speaker at different distances, different angles of arrival, and different self-rotation angles from the radar for experimental analysis and verification, as shown in Figure 7. Figure 7a is the experimental schematic diagram for the speaker and the loudspeaker used to test the playback attack located at different distances from the millimeter-wave radar, Figure 7b is the experimental schematic diagram for the speaker and the speaker located at different arrival angles, and Figure 7c shows the experimental schematic when the speaker and the loudspeaker are at different angles from the radar itself.

aDifferent distances

Since the speaker will use the intelligent voice assistant at different distances in the actual scene, we set 0.2 m, 0.4 m, 0.6 m, 0.8 m, and 1.0 m as the different distances to test the performance of mmSafe, and the experimental results are shown in Figure 8.

Figure 8a,b shows the cumulative distribution functions (CDFs) of the error rate of the verification speaker and the playback attack at different distances from the radar, with the *x*-axis indicating the recognition error rate and the *y*-axis indicating the CDFs percentage. Figure 8a shows the CDFs of the verification speaker, and it can be seen that the highest accuracy is achieved when the distance is 0.2 m; the highest accuracy is achieved when about 84% of the test data have an error rate of less than 10%. The worst performance is achieved when the distance is 1.0 m, when about 64% of the test data have an error rate of less than 30%. Figure 8b shows the CDFs against playback attacks, and it can be seen that about 85% of the test data have an error rate of less than 10% when the speaker is at a distance of 0.2 m from the radar, and about 65% of the test data have an error rate of less than 30% when the distance is of 1.0 m. Figure 8c shows the error rates for verifying the speaker as well as resisting playback attacks at different distances, and it can be seen that the error rates for verifying the speaker as well as resisting playback attacks are greater than 20% when the distance is greater than 0.2 m, and the highest error rates are 27.3% and 26.2%, respectively, when the distance reaches 1 m.

In general, for security privacy issues, the distance range cannot be too large; as the distance becomes larger, the mmSafe performance will also decline, so the experiments chose a distance range within 1 m as the verification area, but also just to ensure security privacy.

bDifferent arrival angles

Since the speaker will use the voice assistant at different arrival angles in the actual scenario, we set different arrival angles of 0°, 30°, and 60° to test the performance of mmSafe, and the experimental results are shown in Figure 9.

Figure 9a,b shows the CDFs when the target under test is located at different radar arrival angles. Figure 9a shows the CDFs for verifying the speaker and it can be seen that, when the speaker is located at a 30° angle relative to the radar, similar to the 0° angle, about 37% of the test data error rate is less than 10% and about 99% of the test data error rate is less than 20%. When the speaker is located at a 60° angle relative to the radar, about 97% of these test data have an error rate of less than 30%. Figure 9b shows the CDFs against playback attacks, and it can be seen that when the loudspeaker is located at a 30° angle relative to the radar, about 42% of the test data error rate is less than 10%, and about 99% of the test data error rate is less than 20%. When the loudspeaker is located at a 60° angle relative to the radar, about 20% of these test data have an error rate of less than 10% and about 81% of the test data have an error rate of less than 20%. Figure 9c shows the error rates when the test target situated at different arrival angles relative to the radar, and it can be seen that the missed detection rates are 12.5% and 9.3% when the test target is at a 30° angle relative to the radar arrival angle, and 16.8% and 12.4% when the test target is at a 60° angle relative to the radar arrival angle.

It can be seen that the performance of the mmSafe system decreases as the angle of arrival between the target under test and the radar becomes larger. Since the IWR1642 radar can transmit a beam at an arrival angle of 90°, the power is so low that it cannot detect the target more accurately. Considering the privacy of the security verification system, the experiment was conducted within a 90° angle.

cDifferent self-rotation angles

Since the speaker will use the voice assistant at different angles in the actual scenario, we set different arrival angles of 0°, 45°, and 90° to test the performance of mmSafe, and the experimental results are shown in Figure 10.

Figure 10a,b shows the CDFs when the target under test faces the millimeter-wave radar at different angles relative to itself. Figure 10a shows the CDFs of the mmSafe system to verify the speaker, and it can be seen that the highest accuracy is achieved when the speaker faces the radar at a 0° angle, where about 84% of the test data have an error rate of less than 10%. The worst performance is achieved when the speaker faces the radar at a 90° angle, where about 57% of the test data have an error rate of less than 20%. Figure 10b shows the CDFs against playback attacks, it can be seen that the highest accuracy is verified when the loudspeaker faces the radar at a 0° angle, where about 85% of the test data have an error rate of less than 10%, and the lowest accuracy is verified when the loudspeaker faces the radar at a 90° angle, where approximately 57% of the test data had an error rate of less than 20%. Figure 10c shows the error rates for verifying the speaker and resisting playback attacks, and it can be seen that the maximum miss detection rates were 17.6% and 16.8%, respectively, when the own angle was 90°.

It can be seen that as its self-rotation angle becomes larger, the performance of the mmSafe system will progressively degrade, and it is difficult for the radar to capture the vibration signal when its self-rotation angle is larger than 90°, so the experiment will be set to the measured target’s self-rotation angle within 90° as the experimental area.

#### 4.2.2. Algorithm Dimension

From the experimental results in Section 4.2.1, we can see that mmSafe can perform best when the speaker is 0.2 m away from the radar and when the arrival angle and the speaker’s own angle are 0°. In order to test the impact of different algorithms on the system’s performance, we set up experiments with different feature parameters and classification methods when the speaker distance from the radar is 0.2 m and the arrival angle and own angle are 0° in this section.

dDifferent feature dimensions

In the biometric feature extraction module of the mmSafe system, the MFCC features of the vibration signal are first obtained, followed by the first-order difference and second-order difference of the MFCC features, and finally the three sets of MFCCs are integrated into the final weighted MFCC features. In order to test the difference between weighted MFCCs and traditional MFCCs, the effect of different feature parameters on the performance of the mmSafe system is set, and the experimental results are shown in Figure 11.

The initial MFCC feature is FEAT are shown in Figure 11a, and it can be seen that the verification accuracy of the mmSafe system for playback attacks is 56.3% and 77.2% for other speakers when this feature is used as the evaluation criterion; defeat is the feature obtained after first-order differencing of feat, and the verification accuracy of the system for playback attacks is 56.3% and 66.4% when this feature is used as the evaluation criterion; “dttfeat” is the feature obtained after second-order differencing of feat, and the verification accuracy of the system for playback attacks is 66.4% when this feature is used as the evaluation criterion. The accuracy of the system is 56.3% for playback attacks and 66.4% for other speakers; “dttffeat” is the feature obtained after second-order differencing of feat, and the accuracy of the system is 55.8% for playback attacks and 65.7% for other speakers when this feature is used as the evaluation criterion. Figure 11b shows the error rates of the system for different feature effects, and it can be seen that when the feature parameter is “feat”, the miss detection rates are 43.7% and 22.8% for verifying speakers and resisting playback attacks, and when the feature parameter is “defeat”, the miss detection rates are 44.7% and 33.6% for verifying speakers and resisting playback attacks, while when the feature parameter is “dttfeat”, the miss detection rates are 44.2% and 34.2% for verifying the speaker and resisting playback attacks. When the first three MFCC features are weighted to obtain the weighted MFCC features (final-MFCC), the system has an accuracy of 94.2% for verifying playback attacks, 93.4% for verifying other speakers, and 5.8% and 6.6% for verifying and resisting playback attacks, respectively.

This shows that the “final-MFCC” has the best performance as the characteristic parameter of the mmSafe system.

eDifferent classifiers

In order to test the performance of different classification methods, we set the effect of different methods on the experimental results. The experimental results are shown in Table 1 and Table 2.

As shown in Table 1 and Table 2, three different classifiers, namely Knn, SVM, and Lostic regression, were used to be compared with WMHS, and it can be seen that the accuracy of the mmSafe system for verifying the speaker and playback attacks was 67.4% and 72.6% when Knn was used, 80.7% and 84.2% when SVM was used as the classifier for verifying the speaker and playback attacks, and 76.2% and 78.4% when logistic regression was used as the classifier for verifying the speaker and playback attacks, respectively. 

This shows that the method performs best when using WMHS as the speaker verification module in the mmSafe system.

#### 4.2.3. Speaker State Dimensions

Since speakers in real-world scenarios will use the voice assistant in different states, to test the impact of the speaker’s state dimension on the system performance, we set the speaker’s state on the performance of the mmSafe system when the speaker is indoors, outdoors, with cover, and experiencing a cold. The laboratory intent is shown in Figure 12a, and the experimental results are plotted in Figure 12b.

Figure 12b shows the verification accuracy of the mmSafe system when the state of the subject is different. It can be seen that the accuracy of the system for verifying the speaker and playback attacks are 78.2% and 85.6% when the subject is located in an outdoor environment, 67.4% and 75.3% when the subject is covered by a mask for verifying, respectively, the speaker and playback attacks. The accuracy rates for verifying the speaker and replaying the attack were 93.4% and 94.2%, respectively, when the subject was indoors, and 84.2% and 86.3%, respectively, when the subject spoke hoarsely due to a cold. Figure 12c shows the verification error rates of the mmSafe system when the speaker is in different states. It can be seen that the error rates for verifying the speaker and playback attacks are 21.8% and 14.4%, respectively, when the subject is located in an outdoor environment; and the error rates for verifying the speaker and playback attacks are 32.6% and 24.7%, respectively, when the subject is covered by a mask. The error rates for verifying the speaker and playback attacks were 6.6% and 5.8%, respectively, when the subjects were indoors, and 15.8% and 13.7% for verifying the speaker and playback attacks, respectively, when the subjects spoke in a raspy voice due to a cold.

As can be seen, the performance of the mmSafe system decreases when the target is covered, still has a high accuracy rate for verifying the speaker when the speaker’s voice changes, and is not affected by the indoor and outdoor environments.

## 5. Conclusions

In this work, we have developed the mmSafe system, which captures the speaker’s vocal cord vibrations through the millimeter-wave radar to extract the vocal print and authenticate the speaker through WMHS. mmSafe can resist playback attacks for speaker verification. In order to obtain a high-quality vocal cord vibration signal in a real-world environment, the system samples the larynx at a specific distance and uses a high-frequency filter to remove the effects of low-frequency noise caused by body movement. To verify the reliability of mmSafe, we set up a simulated attack experiment, using the loudspeaker and “Mi AI” playback to simulate an intrusion into the voice conversation interface. Then, we verified the impact on the system in three different dimensions: speaker space dimension, algorithm dimension, and speaker state dimension. The mmSafe system can achieve a maximum of 93.4% accuracy for subject verification and a 5.8% miss detection rate for playback attacks regardless of the situation. In addition, we also considered RF attacks, as they are known to have a general frequency range of 300 KHz to 300 GHz, so the filters in the mmSafe system are good at filtering signals outside the 90 Hz to 200 Hz range. On the other hand, due to the directionality of millimeter waves, the antenna of the attacker’s device must be placed near the millimeter-wave radar, which can be resisted by the method of bioactivity detection in the literature [29]. Therefore, the system is not only robust against replay attacks. Although the mmSafe system has excellent performance, it still has the following shortcomings:
1In this work, the experimental parameters are set in such a way that the time for parsing the data is not sufficient, and the time set for each frame is too short, resulting in too many frames of speech and thus problems with the processing time;2There is no continuous voice data in this work due to a lack of data richness.

In order to address these shortcomings, we will focus on strengthening these two points in our future work. First, the frame duration in the experimental setup is increased, so that the number of frames in the data is reduced, and the parsing time is relatively reduced. Second, when the data processing time is very fast, richer speech data, such as continuous speech, can be collected, which can lead to higher breakthroughs and higher system robustness in the field of privacy and security verification. In our future work, we will use the ML model for verification using a fusion of two sensors, namely a microphone and a millimeter-wave radar, which will not only enable speech enhancement but also a multimodal verification method with higher accuracy and robustness in harsh environments.

## Figures and Tables

**Figure 1 sensors-22-09309-f001:**
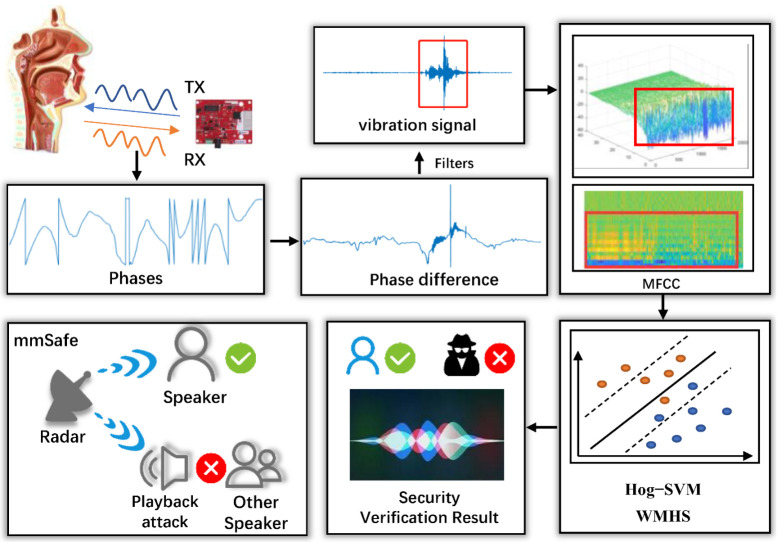
mmSafe system workflow.

**Figure 2 sensors-22-09309-f002:**
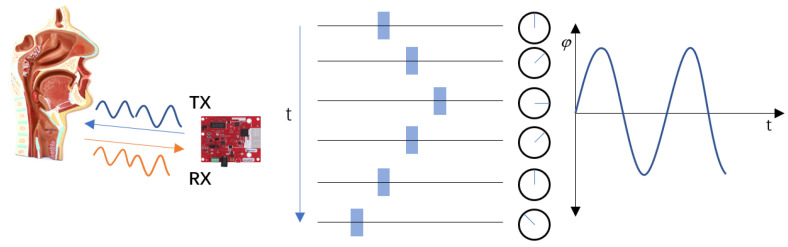
Vibration-induced phase change.

**Figure 3 sensors-22-09309-f003:**
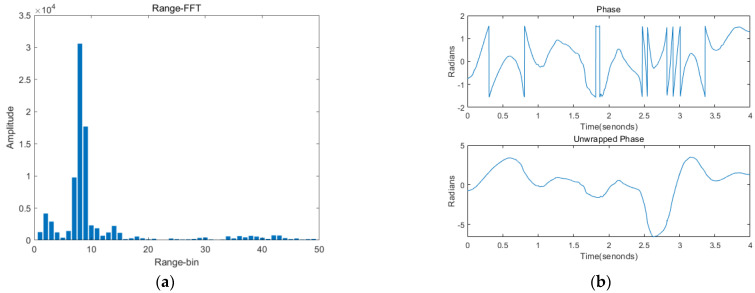
Signal processing: (**a**) Rang-FFT; (**b**) phase; (**c**) phase difference; (**d**) vibration signal.

**Figure 4 sensors-22-09309-f004:**
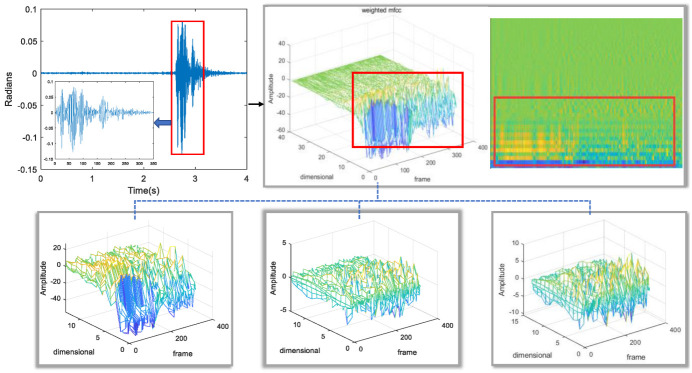
MFCCs.

**Figure 5 sensors-22-09309-f005:**
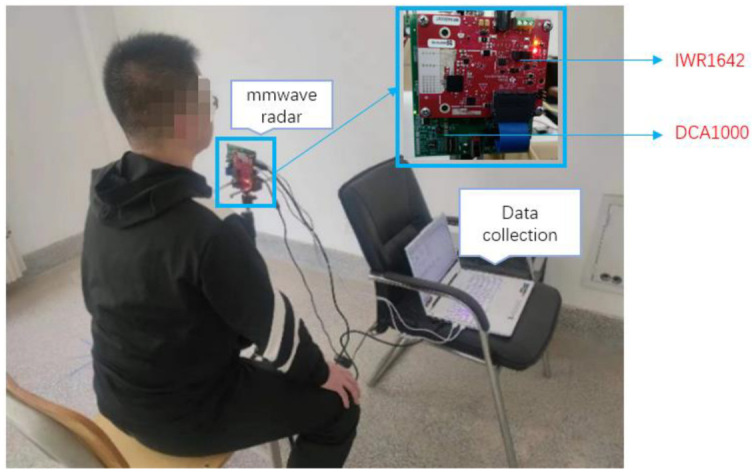
Experiment platform.

**Figure 6 sensors-22-09309-f006:**
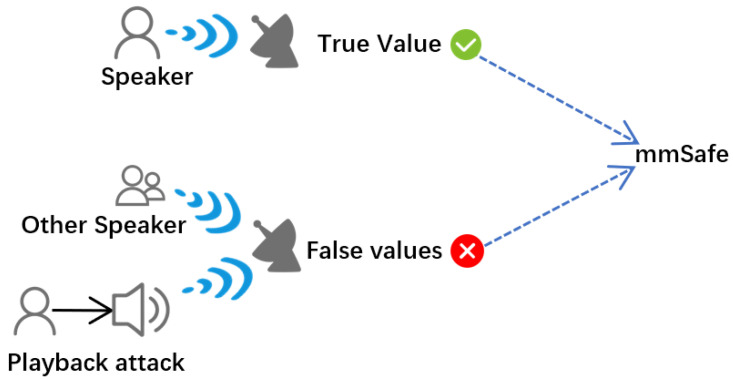
Significance of experimental data.

**Figure 7 sensors-22-09309-f007:**
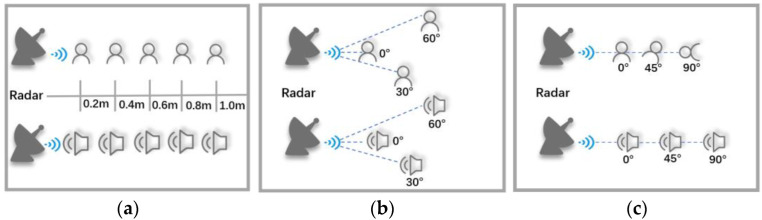
Experimental schematic of the spatial dimension: (**a**) Different distances; (**b**) different arrival angles; (**c**) different self-rotation angles.

**Figure 8 sensors-22-09309-f008:**
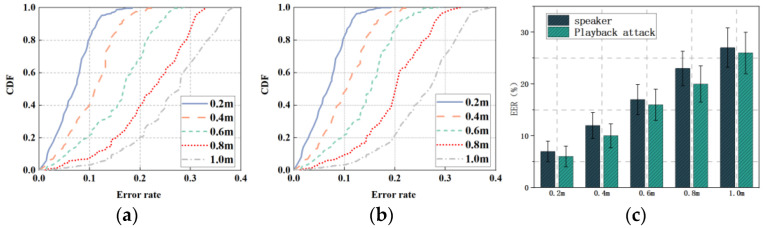
Different distances: (**a**) CDFs (speaker); (**b**) CDFs (playback attack); (**c**) EER.

**Figure 9 sensors-22-09309-f009:**
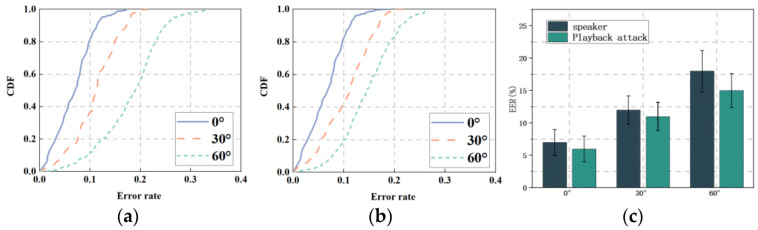
Different arrival angles: (**a**) CDFs (speaker); (**b**) CDFs (playback attack); (**c**) EER.

**Figure 10 sensors-22-09309-f010:**
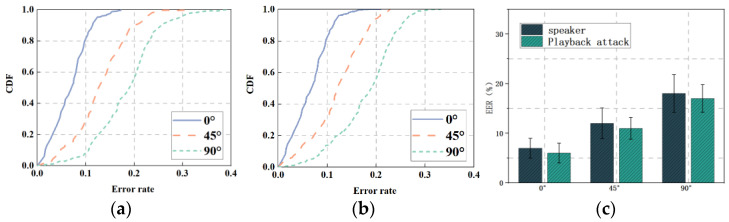
Different self-rotation angles: (**a**) CDFs (speaker); (**b**) CDFs (playback attack); (**c**) EER.

**Figure 11 sensors-22-09309-f011:**
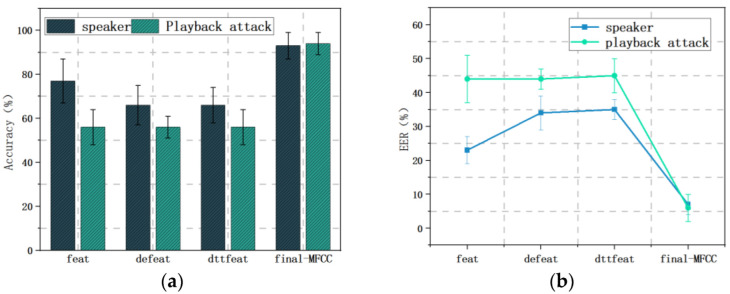
Different features: (**a**) Accuracy; (**b**) EER.

**Figure 12 sensors-22-09309-f012:**
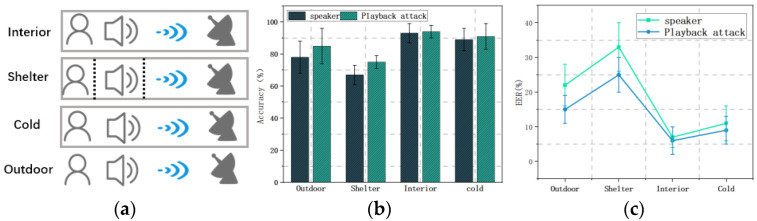
Speaker status dimension: (**a**) Experimental schematic of the speaker state dimension; (**b**) accuracy; (**c**) EER.

**Table 1 sensors-22-09309-t001:** Speaker.

Classification Methods	Accuracy
Knn	67.4%
Lostic	76.2%
SVM	80.7%
**WMHS**	**93.4%**

**Table 2 sensors-22-09309-t002:** Playback attack.

Classification Methods	Accuracy
Knn	72.6%
Lostic	78.4%
SVM	84.2%
**WMHS**	**94.2%**

## Data Availability

Not applicable.

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
