# Peer review of "mmSafe: A Voice Security Verification System Based on Millimeter-Wave Radar"

_sensors, 2022, doi:10.3390/s22239309_

Round 1

Reviewer 1 Report

In this paper, authors have developed the mmSafe system, which captures the speaker's vocal cord vibrations through the millimeter-wave radar to extract the vocal print and authenticate the speaker through WMHS. 

Comments and/or suggestions to the authors. The authors state in the text of the article and in the conclusions that "The mmSafe system can achieve a maximum of 93.4% accuracy for subject verification and a 5.8% miss detection rate for playback attacks regardless of the situation". So the question would be, how do these results look with the works of other authors? Maybe it would be worth adding a few sentences in the conclusions or in the text of the article.

Overall merit. The paper is well organized, has good English language, exact definitions, good graphical representation. The paper covers quite a practical type of design which could be interesting to both industry and the scientific community. I'd suggest accepting this paper with minor revisions after addressing the comments and/or suggestions above.

Reviewer 2 Report

The authors have proposed a mm-wave system for speaker verification. The experimental results are of merit, and the authors have claimed to achieve a  success rate of 93.4% for verification. The authors have also considered the impact of playback attacks. Overall, the methodology and results are intriguing. However, the paper can be improved in terms of clarity, and some additional questions need to be answered. Below are some major comments -  

1/ In section 4.1, what exactly happens in the training or feature extraction phase? In other words, what is the nature of the dataset that is used for learning the features of the authentic speaker? 

2/ Can the performance in section 4.2.1 be improved by learning the features of the speaker from different distances, angles, etc. In other words, is the training set expansive enough?

3/ The results in section 4.2.2 are confusing. How are these results different from those in section 4.2.1? Why are there no results in 4.2.2 for different distances, angles, etc.

4/ The authors have not considered the case where RF attacks can occur. In other words, what if a malicious entity can generate mm-wave signals that affect the performance of the proposed system? Please provide comments on how the proposed system is robust to mm-wave attacks and not just acoustic playback attacks.

5/ Have the authors considered a joint approach (using both microphone and mm-wave signals) for verification? Why not use both types of signals as input to the proposed framework? ML models can accept a variety of signals as input modalities.

Minor comment - Some abbreviations have not been defined (such as SVM). The authors could also include a table or a list of abbreviations as there are many such terms in the paper.
